# Placenta-Derived Mesenchymal-like Adherent Stromal Cells as an Effective Cell Therapy for Cocaine Addiction in a Rat Model

**DOI:** 10.3390/pharmaceutics14071311

**Published:** 2022-06-21

**Authors:** Hilla Pe’er-Nissan, Hadas Ahdoot-Levi, Oshra Betzer, Pnina Shirel Itzhak, Niva Shraga-Heled, Iris Gispan, Menachem Motiei, Arthur Doroshev, Yaakov Anker, Rachela Popovtzer, Racheli Ofir, Gal Yadid

**Affiliations:** 1Neuropharmacology Laboratory, The Mina & Everard Goodman Faculty of Life Sciences, Bar-Ilan University, Ramat Gan 5290002, Israel; peer.hilla@gmail.com (H.P.-N.); hadasahdoot@gmail.com (H.A.-L.); pnina80206@gmail.com (P.S.I.); iris.gispan@biu.ac.il (I.G.); 2The Leslie and Susan Gonda (Goldschmied) Multidisciplinary Brain Research Center, Bar-Ilan University, Ramat Gan 5290002, Israel; oshra.betzer@gmail.com; 3Faculty of Engineering & The Institute of Nanotechnology and Advanced Materials, Bar-Ilan University, Ramat Gan 5290002, Israel; motiei.biu@gmail.com (M.M.); rachela.popovtzer@gmail.com (R.P.); 4Pluristem Therapeutics Inc., Haifa 3508409, Israel; nivas@pluristem.com (N.S.-H.); racheli@pluristem.com (R.O.); 5The Department of Chemical Engineering, Biotechnology and Materials Ariel University, Ariel 40700, Israel; arthurdoroshev5@gmail.com (A.D.); kobia@ariel.ac.il (Y.A.)

**Keywords:** addiction, animal model, neurogenesis, cocaine, drug self-administration, mesenchymal stem cell, cell therapy, intranasal administration, gold nanoparticle cell labeling

## Abstract

Recent research points to mesenchymal stem cells’ potential for treating neurological disorders, especially drug addiction. We examined the longitudinal effect of placenta-derived mesenchymal stromal-like cells (PLX-PAD) in a rat model for cocaine addiction. Sprague–Dawley male rats were trained to self-administer cocaine or saline daily until stable maintenance. Before the extinction phase, PLX-PAD cells were administered by intracerebroventricular or intranasal routes. Neurogenesis was evaluated, as was behavioral monitoring for craving. We labeled the PLX-PAD cells with gold nanoparticles and followed their longitudinal migration in the brain parallel to their infiltration of essential peripheral organs both by micro-CT and by inductively coupled plasma-optical emission spectrometry. Cell locations in the brain were confirmed by immunohistochemistry. We found that PLX-PAD cells attenuated cocaine-seeking behavior through their capacity to migrate to specific mesolimbic regions, homed on the parenchyma in the dentate gyrus of the hippocampus, and restored neurogenesis. We believe that intranasal cell therapy is a safe and effective approach to treating addiction and may offer a novel and efficient approach to rehabilitation.

## 1. Introduction

Drug addiction is the compulsion to consume a drug while losing control over the amount consumed [1]. Drugs like cocaine have a high propensity for reinstatement, occasionally occurring in the short- and prolonged-term abstinence time [2,3]. Therefore, cocaine craving and relapse are crucial challenges in treating cocaine addiction [4,5]. Addiction is a complex brain disease [6] characterized by an impairment in the mesolimbic system, including the prefrontal cortex (PFC), nucleus accumbens (NAc), and the hippocampus; all control two key roles in addiction: motivation and reward behavior. These reward-associated brain regions are co-innervated and, therefore, interesting potential addiction therapy targets [7]. Learned drug-context memories in the hippocampus are enhanced in relation to substance use, such as amphetamine, nicotine, and cocaine. Cocaine, as a strong learning and memory substance stimulant, contributes to the development of maladaptive drug-context association memory, resulting in hippocampal cognitive and plasticity deficits during usage and withdrawal, contributing later to relapse [8].

Neurogenesis in the adult hippocampus forms in its dentate gyrus (DG), creating newly functional neurons that integrate into an existing neural network, thus contributing to the hippocampal plasticity and function [9]. Disruption of neurogenesis in the DG is correlated with exposure to stress and substance use disorder (SUD) [10]. Therefore, it has been suggested that increasing neurogenesis may re-shape memories and decrease drug use behavior and rewarding memories [11]. Numerous studies have demonstrated a neurogenesis decrease in response to addictive drug consumption, such as cocaine [11], amphetamine [12], alcohol [13], and heroin [14], and consequently an increase in the propensity to relapse. In addition, decreasing neurogenesis prior to drug exposure in animals increases vulnerability to drug-seeking behavior [15]. Nevertheless, currently, no FDA-approved treatment for cocaine addiction is effective and safe in maintaining long-term abstinence [16].

Mesenchymal stem cells (MSCs) have emerged as a promising therapeutic modality for a variety of central nervous system disorders, including neurodegenerative disorders (e.g., Alzheimer’s [17] and Parkinson’s [18]), neuropsychiatric diseases (e.g., depression [19,20] and schizophrenia [21]), and peripheral pathologies [22]. MSCs migrate and home to injured regions, including the brain [23], in response to pathological sites’ pro-inflammatory signaling, which activates MSCs’ migratory tropism and adhesion capacity toward the distant, damaged tissue [24]. MSCs secrete varied factors [19,25,26], among them neurotrophic factors that reduce inflammation and prevent oxidative stress [27], which are linked to cell survival, connectivity, and neurogenesis [20,28]. As such, MSCs hold a potential therapy for SUD. Specifically, the MSCs can migrate to pathologic brain regions [23] and promote neurogenesis and cell survival [26], which are impaired in SUD [29].

Despite the potential of using MSC therapy to treat SUD, studies in rodent models of addiction are scarce [30,31]. Israel et al. (2017) showed that intracerebroventricular (ICV) administration of MSCs inhibited alcohol consumption and relapse in rats [30]. However, the highly invasive nature of ICV administration makes it both impractical and unsafe for clinical use. Quintanilla et al. (2019) showed that intranasal (IN) administration of the MSC secretome inhibited alcohol relapse binge and chronic alcohol and nicotine self-administration [31]. However, the secretome contained only a limited number of factors derived from MSCs pre-incubation with inflammatory-reached media. Once these factors reached the target region and actioned on it, they dissolved. Hence, for adequate results, the secretome was administrated five-time doses periodically along alcohol and nicotine self-administration and the deprivation phases. It is conceivable, then, that treatment with MSCs, rather than their restricted version as secretome, may be a more efficient therapy due to their viable, dynamic long-term-therapeutic features.

In the current work, we evaluate the potential of IN administration of an innovative MSC-like therapy, PLX-PAD, to treat cocaine addiction. The PLX-PAD cells are placenta-derived MSC-like cells with immunomodulatory and regenerative properties similar to those of MSCs [32]. These such capacities of PLX-PAD cells have been shown to improve blood flow to the ischemic limb in mice [33] and in severe critical limb ischemia patients [34]. Moreover, they were shown to induce muscle regeneration and improve the strength and volume of injured skeletal muscle [35]. Like other MSCs, PLX-PAD cells are hypoimmunogenic: they express low levels of major histocompatibility complex (MHC) class I, do not express MHC class II (HLA-DR) or co-stimulatory molecules (CD80, CD86, CD40), and do not induce activation of allogeneic lymphocytes [36]. Importantly, they have the capacity to be available off the shelf. However, unlike MSCs, PLX-PAD cells have limited proliferation and minimal differentiation capabilities [32,37], thus avoiding some of the shortcomings of classic MSCs, including tumorigenesis risk and concerns regarding final cell fate post-transplantation [38]. We hypothesize that PLX-PAD cells migrate to and promote neurogenesis in brain regions impaired by cocaine addiction, thereby decreasing cocaine craving and relapse. To evaluate this hypothesis, we tested the short- and long-term effects of PLX-PAD cells on cocaine-seeking behavior in the self-administration rat model. We tracked the cell migration using theranostic gold nanoparticle cell labeling in vivo and ex vivo. 

## 2. Materials and Methods

### 2.1. Cells

PLX-PAD is an allogeneic ex vivo placental-expanded adherent stromal cell product whose manufacturing procedure and characterization have been previously described [35]. Briefly, the cells are derived from a full-term human placenta following a cesarean section and are expanded using plastic adherence on tissue culture dishes, followed by three-dimensional growth on carriers in a bioreactor. PLX-PAD cells are aseptically transferred to cryogenic vials at a concentration of 20 × 10^6^ cells/mL in a mixture containing 10% dimethyl sulfoxide, 5% human albumin, and Plasma-Lyte (Baxter Healthcare Ltd., Toongabbie, NSW, Australia). Storage takes place in gas phase liquid nitrogen at temperatures below −150 °C. The required amount of PLX-PAD was thawed in a heated water bath (37 °C) immediately prior to injection.

### 2.2. Animals

A total of 86 male Sprague–Dawley rats (250–280 g) were maintained on a reverse 12–12-h dark-light cycle with free access to food and water. All experimental procedures were approved by the Animal Care and Use Committee of Bar Ilan University, Ramat Gan, and were performed in accordance with National Institutes of Health guidelines.

### 2.3. Catheter and Guide Cannula Implantation

All rats were anesthetized with ketamine hydrochloride [100 mg/kg, intraperitoneally (i.p.)] and xylazine (10 mg/kg, i.p.), then implanted with intravenous Silastic catheters (Dow Corning, Midland, MI) into the right jugular vein. The catheter was secured to the vein with silk sutures and was passed subcutaneously to the top of the skull, where it exited into a connector (a modified 22-gauge cannula; model: C313G-5UP-PLA-8IC313G5UPXC, web link: https://catalog.p1tec.com/Search.php?SEARCHSTRING=C313G-5UP, accessed on 25 May 2022, Plastics One, Roanoke, VA, USA) that was mounted to the skull with MX-80 screws (Small Parts, Inc., Miami Lakes, FL, USA) and dental cement (Yates and Bird, Chicago, IL, USA). Catheters were used for self-administration of cocaine or saline.

Rats that were to undergo intracerebroventricular administration were placed in a stereotaxic apparatus (David Kopf Instruments). A hole was drilled through the skull, and a guide cannula was implanted into the left ventricle of the brain (anterior −0.8, lateral 1.5, ventral −4.0 mm from the bregma) and secured to the skull with dental cement. The guide cannula was used to administer PLX-PAD cells or cerebrospinal fluid (CSF) (vehicle) intracerebroventricular.

### 2.4. Cocaine Self-Administration

Rats were trained to self-administer cocaine (cocaine was obtained from the National Institutes of Drug Abuse, North Bethesda, MD, USA) as previously described [39] under an FR-1 schedule of reinforcement for 13–14 days until reaching stable maintenance of drug intake, as follows. 5 days after catheterization, rats were transferred to operant conditioning chambers (Med-Associates, Inc.; St. Albans, VT, USA) for one hour daily during their dark cycle. Each self-administration chamber (30 × 25 × 22 cm) had two levers, active and inactive, located 9 cm above the chamber’s floor. The self-administration chambers and the computer interface were built locally and controlled by a computer program written by Steve Cabilio (Concordia University, Montreal, PQ, Canada; steve.cabilio@concordia.ca). The program is available via Med Associates Inc. (St. Albans, VT, USA.) https://www.med-associates.com, accessed on 25 May 2022. An active lever press generated a cocaine infusion (1.5 mg/kg in saline, 0.13 mL total volume, over 20 s) through the IV catheter connected to an infusion pump. During cocaine infusion, a light located above the active lever was lit for 20 s. During the 20 s infusions, active lever presses were recorded, but no additional cocaine reinforcement was provided. Presses on the inactive lever did not activate the infusion pump or light. The number of active lever presses, infusions, and inactive lever presses was recorded. Rats were returned to their home cages at the end of the daily session. Sham groups underwent the same self-administration protocol with saline instead of cocaine. PLX-PAD or vehicle (or gold nanoparticles in navigation experiments, below) was administered intranasal or intracerebroventricular 24 h after the last self-administration session.

### 2.5. Extinction Phase and Reinstatement Test (Relapse)

During the extinction period, rats were placed in the operant chamber for 60-min daily sessions with no cocaine or saline access (active and inactive lever presses were recorded but had no effect). Rats underwent drug-extinction training for 10 days until the number of active lever presses decreased significantly from the first day of drug-extinction training and was, therefore, assumed to be non-reinforced by cocaine. The rats used in navigation experiments were sacrificed during the extinction period.

The reinstatement test to assess cocaine craving during relapse was performed 24 h after completion of the last extinction session (or 28 days after BrdU injection in neurogenesis experiments), as follows. Cocaine-trained rats were primed with a single cocaine injection (10 mg/kg i.p.), then placed in the self-administration (operant) chamber for 1 h, with no cocaine dispensed; lever presses were monitored but had no effect. Sham groups underwent the same relapse protocol with saline instead of cocaine. Rats completing the relapse test were decapitated, and their brains were removed.

### 2.6. Intracerebroventricular Injection

In intracerebroventricular administration, 0.75 × 10^6^ viable PLX-PAD cells in 10 µL within 1 h of thawing (or vehicle: 10 µL CSF (Sigma-Aldrich, Rehovot, Israel)) were injected at 1 µL per min, using a 26-gauge Hamilton syringe connected to a polyethylene tube attached to the guide cannula.

### 2.7. Intranasal Administration

Rats that were to undergo intranasal (IN) administration were anesthetized either by i.p. injection of 100 mg/kg ketamine and 10 mg/kg xylazine or by placement for 20 min into an anesthesia chamber: the size of the normal home cage containing clean bedding with 2% isoflurane and 98% air. 1 × 10^6^ PLX-PAD cells in 50 µL within 1 h from thawing (or vehicle: 50 µL PlasmaLyte) were administered IN using the Impel Rat Intranasal Catheter Device (Neuropharma, Israel) using SP2 settings, half into each nostril. The catheter tube was inserted into the nostril (5 mm deep) using a guide catheter. Cells were slowly released inside each nostril (25 µL/15 min), ensuring that they did not touch the nasal mucus. To enable a maximum number of cells to pass without leaking, the rat’s head was held tilted up at 15 degrees.

### 2.8. Tracking PLX-PAD Cell Navigation Using Gold Nanoparticle Labeling

To track the movement of PLX-PAD cells in the brain and body, cocaine- and saline-trained rats were treated with gold nanoparticle (GNP)-labeled PLX-PAD cells (“nPAD”) or with free GNPs (fGNPs), 24 h after completing the self-administration phase. Extinction training commenced 24 h after nPAD or fGNP administration. At 24 h, 72 h, and 1 week after nPAD or fGNP administration, a subset of the rats was subjected to in vivo CT followed by sacrifice and GNP quantification using inductively coupled plasma optical emission spectrometry (ICP-OES), as described below.

### 2.9. Gold Nanoparticle Labeling of PLX-PAD Cells

PLX-PAD cells were labeled with gold nanoparticles (GNPs), as in previous work [40]. A total of 20 nm GNPs were synthesized, coated with PEG7, and covalently conjugated to glucose to enable uptake by cells, as described previously. GNPs labeling was shown previously to have no effect on cell function and viability and the cells remained biologically and functionally active [41].

### 2.10. Cell Uploading with Gold Nanoparticle

GNPs synthesizing and preparation and MSCs labeling with GNPs as described previously [40]. PLX-PAD cells were cultured in 5 mL glucose-free DMEM medium containing 5% FCS, 0.5% penicillin and 0.5% glutamine. Cells were centrifuged, and a saline solution containing GNPs (30 mg/mL) was added in excess. The cells were then incubated at 37 °C for 2 h. After incubation, the cells were centrifuged twice (7 min at 1000 rpm) to wash out unbound nanoparticles. Then, we examined the feasibility of the uptake of GNPs by the cells; PLX-PAD cells were incubated with the GNPs for 2 h, and the average amount of GNPs uptake was analyzed using inductively coupled plasma optical emission spectrometry, and as will be further described, this was found to be 1.04 million particles per cell (std: 0.12) (8.396 × 10^−8^ mg gold per cell).

### 2.11. In Vivo Micro Computed Tomography Scans

In vivo computed tomography (CT) scans were conducted as previously [41] at 24 h, 72 h, and 1 week after IN administration of PLX-PAD. Animals were scanned in vivo with a micro-CT scanner (Skyscan High Resolution Model 1176, Bruker micro-CT, Kontich, Belgium) at a nominal resolution of 35 μm, using a 0.5 mm aluminum filter and an applied X-ray tube voltage of 60 kV, with current source 350 μA (detection limit ~1000 GNP-labeled cells).

### 2.12. Inductively Coupled Plasma Optical Emission Spectrometry

PLX-PAD cells present in different brain regions and peripheral organs were quantified by measuring GNP content using inductively coupled plasma optical emission spectrometry (ICP-OES). To prepare the samples for analysis, excised tissues were acidified in aqua regia solution (1:3 molar ratio of nitric acid: hydrochloric acid), then heated to 110 °C for 24 h, filtered through a 0.22 µm nylon syringe filter, and diluted in half with deuterium-depleted water. The analyzer was recalibrated each time it was turned on, using a series of three standard solutions (0.5, 1, and 2 mg/L gold) prepared from a 1000 mg/L gold standard solution (Merck). Gold concentration was measured with a Liberty 110 ICP-OES spectrometer (Varian, Australia) for the Au 242.795 emission wavelength. The measurement conditions included an argon plasma flow rate of 15 L/min with 1.2 kW RF power; the auxiliary flow was 1.5 L/min with 200 kPa nebulizer pressure.

The number of cells per region was calculated by the amount of gold per region divided by the number of GNPs taken up by the cells (see GNP labeling section).

### 2.13. Identifying PLX-PAD Cell Navigation with Immunohistochemistry

Four rats underwent 14 days of cocaine self-administration training, followed by administration of unlabeled PLX-PAD 24 h after the last self-administration session. After one week of extinction training that began 24 h after PLX-PAD administration, rats were anesthetized. Phosphate buffer (PBS) perfusion was performed to the left ventricle in the heart until the blood washed out through the right atrium, then 4% paraformaldehyde was injected in the same manner for brain fixation. Brains were removed, post-fixed overnight, and equilibrated in phosphate-buffered 30% sucrose. Free-floating, 40-μm-thick coronal hippocampal sections were collected on a freezing cryostat and stored at 4 °C before immunohistochemistry.

To visualize cell navigation along the brain’s parenchyma, brain slices, including the dentate gyrus, were stained with DAPI and anti-Ku80, a specific human nuclear protein antibody that recognizes PLX-PAD cells. Specifically, slices were washed in PBS, treated with blocking solution (1 h), and then incubated overnight with rabbit anti-Ku80 (1:250; Abcam, # ab80592) as a primary antibody for PLX-PAD cells. Slices were then incubated with secondary antibody (1:200 Alexa Fluor488 Anti-Rabbit (Invitrogen, Hillsboro, Oregon, USA), for 1 h) and with DAPI (0.0005 mg/mL, 2 min) to visualize cell nuclei.

### 2.14. Quantification of Neurogenesis

Neurogenesis was assessed as in our previous work [11]. As previously, rats were trained to self-administer cocaine or saline for 14 days until reaching stable maintenance levels. Twenty-four hours after the last self-administration session, cocaine-trained rats were treated IN with PLX-PAD or vehicle. Twenty-four hours after IN treatment in cocaine-trained rats, all rats were injected i.p. with three injections of BrdU (Sigma-Aldrich; 50 mg/kg body weight). Then, 10-day extinction training began 24 h after the first BrdU injection. Twenty-eight days after the BrdU injections, rats were subjected to the relapse test, then euthanized and perfused transcranially with PBS, then with 4% paraformaldehyde. Coronal sections of the dorsal DG (40 µm thick) were collected on a freezing cryostat. Every fifth section (a total of five sections, 200 µm apart) was taken for immunohistochemical analysis and was stored free-floating in PBS containing sodium azide (1%) at 4 °C.

For immunohistochemical analysis of BrdU and NeuN (a marker of adult neurons), tissue sections were washed with PBS, incubated in 2N HCl at 37 °C for 30 min, then blocked for 1 h with a blocking solution (PBS containing 20% normal horse serum and 0.5% Triton X-100). The tissue sections were stained overnight with rat anti-BrdU (1:200; Oxford Biotechnology, Kidlington, Oxfordshire, UK) and mouse anti-NeuN (1:200; Chemicon, Temecula, CA, USA) as primary antibodies. Secondary antibodies used were Cy-3-conjugated donkey anti-rat (1:200; Jackson ImmunoResearch, West Grove, PA, USA) and Cy-2 donkey anti-mouse (1:200; Jackson ImmunoResearch). To evaluate neurogenesis, we counted the cells that were co-labeled with BrdU and NeuN in five coronal sections per rat brain that were stained and mounted on coded slides (blind to the evaluator), using confocal microscopy (Leica inverted SP8 scanning confocal microscope, Leica Microsystems, Mannheim, Germany; driven by LASX software and using an HC PL APO 20×/0.75 objective). To estimate the total number of co-labeled cells per DG, the total number of BrdU-NeuN-positive cells counted in the selected coronal sections from each brain was multiplied by the volume index (the ratio between the volume of the DG and the total combined volume of the selected sections). Images of the dorsal dentate gyrus were captured by a confocal microscope at 10× and 20× magnification.

## 3. Results

### 3.1. Intracerebroventricular (ICV) and Intranasal (IN) Administration of PLX-PAD Cells Alleviates Cocaine Extinction and Relapse of Usage Behavior

To investigate whether PLX-PAD could serve as a potential therapeutic strategy for cocaine addiction, we trained rats to self-administer cocaine or saline (sham) until maintenance dose consumption was attained by Day 13, then administered PLX-PAD or vehicle ICV or IN on Day 14 (see Figure 1 for design and groups). Cocaine craving was assessed by active lever presses recorded in the operant chamber during the self-administration phase (Days 1–13), extinction phase (Days 15–24; no cocaine available), and the relapse test on Day 25, when cocaine- and saline-treated rats were reinstated with cocaine (10 mg/kg, i.p.) or saline, respectively, prior to placement in the operant chamber with no additional cocaine available.

We assessed the rats’ cocaine-seeking behavior during the self-administration and extinction phases (Figure 1b). Two-way ANOVA with repeated measures of active lever presses over Days 1–24 in the five groups revealed significant day Factor [F(22, 1022) = 48.29, *p* < 0.0001], group Factor [F(4, 1022) = 23.59, *p* < 0.0001] and group × day interaction [F(88, 1022) = 4.590, *p* < 0.0001]; post hoc analysis was performed using Bonferroni’s multiple comparison tests. On Day 15, the first day of the extinction phase, cocaine-trained rats treated ICV and IN with vehicle exhibited significantly more (*p* < 0.0001) active lever presses than the ICV and IN PLX-PAD-treated groups, respectively, indicating higher craving for the drug. No significant difference was observed between PLX-PAD treatment delivered IN and ICV.

Then, we assessed the rats’ propensity to relapse, assessed as cocaine-seeking behavior (active lever presses) after cocaine reinstatement on Day 25 (Figure 1c). One-way ANOVA of active lever presses on Day 25 showed a significant difference between groups [F(4, 45) = 26.67, *p* < 0.0001]; post hoc analysis was performed using Tukey’s multiple comparison tests. Cocaine-trained rats treated with the vehicle by either administration route exhibited significantly more (*p* < 0.0001) active lever presses than the ICV and IN PLX-PAD-treated group. Dramatically, the number of active lever presses in cocaine-trained rats treated with PLX-PAD by either administration method was not significantly different from that in sham rats and was significantly lower than that in the corresponding vehicle-treated group (*p* < 0.0001 for IN, *p* < 0.01 for ICV). As on the first day of the extinction phase, no significant difference was observed between PLX-PAD treatment delivered IN and ICV in the relapse test. Since IN delivery of PLX-PAD cells appeared to be at least as effective as ICV administration while being much less invasive, we continued this study using IN administration only.

### 3.2. PLX-PAD Cells Can Navigate to Specific Pathological Addiction-Associated Brain Regions

Next, we examined the ability of PLX-PAD cells administered IN to decrease cocaine craving by penetrating the brain from the nasal cavity and navigating to specific brain regions. To enable cell tracking, we labeled the cells to gold nanoparticles (GNPs), as previously described. Rats were trained to self-administer cocaine or saline (sham) as above. After maintenance, drug consumption was reached (by Day 14), and GNP-labeled PLX-PAD cells (“nPAD”) were administered IN on Day 15 (see Figure 2a for design and groups). As a control, additional cocaine-trained rats were administered free GNPs (fGNPs) instead of nPAD, and sham rats were treated IN with GNP-labeled PLX-PAD cells (‘Sham-nPAD’). Twenty-four hours after treatment, on Day 16, extinction training began. At 72 h post-treatment (Day 18), the Coc-fGNP and Sham-nPAD groups and a subset of the Coc-nPAD group underwent in vivo CT scanning followed by sacrificing and analysis of GNP content in various brain regions, lungs, and liver, by inductively coupled plasma optical emission spectrometry (ICP-OES). Subsets of the Coc-nPAD group underwent the same procedure at 24 h and 1-week post-treatment. The number of nPAD in each region or organ was estimated from the number of the GNPs (see Cell uploading with GNP parts in Methods).

To examine if the nPAD navigation and accumulation 72 h post-treatment was in parallel to pathology, we compared cocaine self-trained rats (‘Cocaine+nPAD’ or ‘Cocaine+fGNP’) and saline self-trained rats (‘Sham-nPAD’) (n = 4–7 per group) (Figure 2b). Two-way ANOVA of the number of nPAD equivalents in each brain region in the three treatment groups revealed a significant main effect of treatment group [F (2, 98) = 20.77, *p* < 0.0001], brain region [F (6, 98) = 8.096, *p* < 0.0001], and treatment group × brain region [F (12, 98) = 2.777, *p* = 0.0027]; post hoc analysis was performed using Tukey’s multiple comparison tests within each brain region. In all brain regions, more GNPs were observed in the nPAD-treated groups than in the fGNP-treated group. These differences were significant in the ventricles (Coc-fGNP vs. Coc-nPAD *p* < 0.0001, vs. Sham-nPAD *p* < 0.01) and reward-associated brain regions (Coc-fGNP vs. Coc-nPAD: *p* < 0.0001 in PFC, *p* < 0.01 in NAc and DG). The significant differences between the fGNP- and nPAD-treated groups confirm that the GNPs observed in the different brain regions in nPAD-treated groups were due to PLX-PAD migration rather than accumulation of unbound GNPs.

#### 3.2.1. IN-Administrated PLX-PAD Cells Reach the Brain Mainly through the Olfactory Route

The IN administration is carried out by two major pathways that differ from each other by travel distance: the olfactory route to the olfactory bulb (OB) and the trigeminal nerves to the brain stem (BS) (which is longer) [42]. To verify that the nPAD reached the brain via the intranasal route, we evaluated the number of GNPs in the OB and the BS in the nPAD- and fGNP-treated groups. The low number of GNPs indicates that nPADs and fGNPs that reached the OB and BS from the nasal cavity did not accumulate but moved on to other brain regions. Notably, no significant difference was found in nPAD accumulation between cocaine-trained and sham rats in the OB region, confirming the trajectory of the cells to the brain via the nasal cavity. Comparisons of the nPAD number between cocaine-trained and sham rats at 72 h revealed a significantly higher (*p* < 0.01) number in cocaine-trained than in sham rats in the PFC and the NAc, but no difference in the DG (discussed below) and the ventricles. Altogether, these findings show that cocaine-induced pathology increases PLX-PAD migration to these regions and the cells have the ability and accessibility to navigate there within the brain from the ventricles via the CSF. Surprisingly, more cells were present in the reward-associated brain regions than in the reward-unassociated brain regions. We assumed that increased activation of cells in these regions [43,44] would increase blood flow. This could contribute to cell migration and distribution. Accordingly, not all cells had completed their migration along the lengthy trigeminal nerves to the brain [42].

#### 3.2.2. PLX-PAD Cells Reach the Brain in the First 24 h Post-IN Administration

To examine the PLX-PAD migration over time, we examined the GNP content in cocaine-trained nPAD-treated rats across the three evaluated time points (24 h, 72 h, and 1-week post-IN treatment) (Figure 2c). As reward-associated and unassociated regions exhibited very different GNP counts at 72 h (Figure 2b), each set was pooled to overview differences between the groups. The GNP content was also assessed in the non-targeted organs lungs and liver (presented pooled) to evaluate PLX-PAD leaking and accumulation in these organs. Two-way ANOVA of nPAD number in the groups reward-associated, reward-unassociated, and non-targeted organs across the three time points revealed a significant main effect of the groups [F (1.004, 20.08) = 71.98, *p* < 0.0001] but not of the time point; post hoc analysis was performed by Tukey’s multiple comparison test. Non-targeted organs showed minimal nPAD accumulation, significantly lower than in reward-associated brain regions (*p* < 0.01, *p* < 0.001, *p* < 0.0001) and reward-unassociated brain regions (*p* < 0.01), supporting a favorable biodistribution profile of PLX-PAD with minimal accumulation in non-target organs. It is noteworthy that MSCs and PLX-PAD cells in the liver and lungs are not associated with toxicity effects [45,46]. The reward-associated brain regions showed significantly more (*p* < 0.001) nPADs than reward-unassociated regions at 24 h, and a similar effect (*p* < 0.0001) was seen at the 72 h and 1 week (*p* < 0.001) time points as well (Figure 2c). No significant differences between the time points were observed in any pooled groups, suggesting that most PLX-PAD cells migrate to their final destination within the first 24 h.

#### 3.2.3. PLX-PAD Cells’ Particular Kinetic Pattern in the Dentate Gyrus

To investigate the PLX-PAD dynamic migration pattern, we examined the nPAD kinetics individually within each reward-associated brain region in cocaine-trained rats (Figure 2d). Two-way ANOVA of nPAD number in the three brain regions over the three time points showed a significant main effect of time [F (2,42) = 4.242, *p* = 0.0210]; post hoc analysis was performed by Tukey’s multiple comparison tests. No significant changes over time were observed in the PFC or NAc, corresponding to the pooled results shown in Figure 2c. The DG, however, showed a particular kinetic pattern evident by a significant decrease in nPAD number between 24 h and 72 h post-treatment (*p* < 0.05). This migration of nPAD away from the DG by 72 h in cocaine-trained rats may explain why no significant difference was observed between nPAD number in the DGs of cocaine-trained and sham rats at this time in Figure 2b. The nPAD kinetics in the DG measured by ICP-OES (Figure 2d) is supported by representative CT scans (Figure 3a), which visually show the decrease in nPADs in the DG between the 24 h and 72 h time points. Like the PFC and NAc, individual reward-unassociated brain regions and untargeted organs did not show any significant differences in nPAD number between the three time points in cocaine-trained animals (not shown).

Taken together, the evidence presented in Figure 2b–d indicates migration of the PLX-PAD cells to all three reward-associated brain regions. Panel (b) shows that in all the reward-associated-brain regions tested (PFC, NAc, and DG), gold nanoparticles are highly accumulated in rats trained to self-administer cocaine when PLX-PAD cells were conjugated to gold nanoparticles, and migration is low in saline-trained rats. Sole treatment with gold nanoparticles barely yielded migration (‘coc-fGNP’). Moreover, panel (d) shows that PLX-PAD cells are in dynamic movement throughout time. Considering panel (b), had it been only gold nanoparticles (without the PLX-PAD cells), this special effect could not have been received. Hence, we can strongly confirm that PLX-PAD cells migrated to the brain and to selective regions.

Immunohistochemistry was performed to confirm that GNPs observed at one week in the Coc-nPAD group in the DG indicated live nPAD cells (Figure 3b). The DG region was elected for staining since it is affected by cocaine consumption that damages neurogenesis in this region [11,44], which is a main focus of this study. Representative images of the DG show co-localization of the Ku80. The DAPI stains indicate that the PLX-PAD cells were indeed present in the DG and alive 1-week post-IN treatment.

In summary, PLX-PAD cells were barely accumulated in addiction-related brain regions of sham rats. Therefore, it can be concluded that cell accumulation in pathological regions is due to their chemotactic therapeutic properties. Surprisingly, 1-week post-IN, only a minor number of PLX-PAD cells was found in the OB and the BS, the first brain regions that cells reach when migrating from the nasal cavity. This finding strengthens the conclusion that the vast majority of the cells safely migrate to pathological brain regions, where they enhance tissue and neuronal repair. Furthermore, PLX-PAD cells were evenly accumulated in the ventricles both in the cocaine-trained rats and the sham groups, confirming that in both treated groups, the cells had the ability as well as the accessibility to navigate via CSF within the brain to addiction-related areas.

### 3.3. PLX-PAD Cells Induce Renewal of Hippocampal Neuronal Cells (Neurogenesis)

As MSCs impact neurogenesis [20,21,28], and the nPAD kinetics observed in the DG between 24 h and 72 h (Figure 2d) suggested that PLX-PAD may have a significant role in the DG in the first 24 h post-treatment, we hypothesized that PLX-PAD might stimulate neurogenesis in the DG during the first 24 h after its administration. To investigate whether this could be related to neurogenesis in the DG, adult neurogenesis in PLX-PAD-treated and un-treated rats was evaluated by labeling with the neurogenesis marker BrdU and staining for BrdU and NeuN, a marker of adult neurons. Rats were trained to self-administer cocaine or saline (until the achievement of stable maintenance by Day 14), then treated IN with PLX-PAD or vehicle on Day 15 (see Figure 4a for design). On Day 16, all rats were injected with the neurogenesis marker BrdU (3 i.p. injections at 4 h intervals; see Methods). Rats underwent 10-day extinction training on Days 17–26. Then, on Day 44 (28 days after BrdU injection), rats were subjected to the relapse test (described above), after which they were sacrificed, brains were excised, and DG sections were stained for BrdU and NeuN.

A major obstacle in treating recurrent relapse to drug use lies in the permanent neuroplasticity caused by drug use. Hence, in light of the PLX-PAD cells’ ability to migrate to mesolimbic brain regions such as the DG, we further investigated the PLX-PAD cells’ effect on neuroplasticity using neurogenesis methods. Neurogenesis was quantified (Figure 4c) in each group of rats by counting the number of BrdU-NeuN co-labeled cells in five coronal sections per brain and scaling by a volume index to estimate the number of co-labeled cells in the DG (see Methods) (Figure 4b). One-way ANOVA of total BrdU-NeuN co-labeled cells in the DG showed a significant difference between groups [F(2,9) = 5.114, *p* = 0.0328]; post hoc analysis was performed using Tukey’s multiple comparison tests. As expected, the vehicle-treated cocaine-trained group exhibited impaired neurogenesis (fewer co-labeled cells) compared to the saline-trained (sham) group. Treatment of cocaine-trained animals with PLX-PAD significantly raised neurogenesis levels compared to the vehicle (*p* < 0.05), restoring neurogenesis to that observed in the sham group (no significant difference from the sham group).

Lastly, we examined the effect of PLX-PAD IN administration on reinstatement. One-way ANOVA of the number of lever presses showed a significant difference between the groups [F(2, 14) = 9.234, *p* = 0.0028] (Figure 4d). Post hoc analysis with Tukey’s multiple comparison tests yielded results similar to those of the relapse test performed on Day 25 (Figure 1c): PLX-PAD treatment significantly decreased (*p* < 0.05) the number of active lever presses compared to the vehicle-treated cocaine-trained rats, restoring the number of lever presses to that observed in the sham group (no significant difference of sham). To determine whether the restoration of neurogenesis correlated with decreased cocaine craving, we correlated the number of active lever presses during the relapse test with BrdU-NeuN co-labeled cells observed in the same rats (Figure 4e). A statistically significant correlation was found (Spearman correlation coefficient r = −0.6063, *p* = 0.0366), suggesting that the restored neurogenesis may have contributed to the decreased cocaine craving. Therefore, IN PLX-PAD treatment was able to rescue the decreased neurogenesis caused by cocaine training and attenuate cocaine craving. Hence, although cocaine training decreased neural survival in the DG during drug consumption and withdrawal, IN administration with PLX-PAD cells increased neural survival and attenuated cocaine craving.

## 4. Discussion

Craving and relapse are major factors in treating cocaine addiction [4], yet current treatments lack a reliable and efficient longitudinal effect [16]. Here, for the first time, we demonstrated the use of the placenta-derived MSC-like cell therapy PLX-PAD to treat cocaine addiction in an animal model. We demonstrated that PLX-PAD significantly decreased cocaine craving and cocaine-seeking behavior during the first day of extinction and reinstatement. Notably, the non-invasive IN administration was as effective as the highly invasive ICV administration, and the relapse test showed a longitudinal effect of PLX-PAD treatment. Moreover, our study shows that this effect is associated with the restoration of neurogenesis in the DG. This is in line with our previous study [20], which shows that MSCs alleviate depressive-like performance in rats by migrating to the hippocampus and promoting neurogenesis.

An added value to our findings is the labeling MSCs to GNPs that we used for tracking the cells’ final location in the brain. This method allows real-time in vivo detection of the exact location and the dynamic migration trajectory of the cells, as well as following up on the progress of tissue repair [47]. Our findings showed that IN-administered PLX-PAD cells navigated to the reward-associated brain regions PFC, NAc, and DG in cocaine-trained animals. Importantly, we demonstrated a unique kinetic pattern of the cells in the DG. PLX-PAD presence in the DG in the first 24 h post-IN administration entails their contribution to the neurogenesis restoration immediately upon arrival. It is intriguing to speculate that by 72 h, the cells have already migrated to the next needed destination in the brain. Overall, our findings support studies showing the MSC cells’ property to navigate and home on specific pathological brain regions, promoting neural survival and neurogenesis [20].

This study emphasizes the unique therapeutic characteristics of MSC therapy—a chronic and progressive therapy that instigates a healing effect—as opposed to the traditional pharmacological existing therapies, which attempt to repair cocaine addiction through its symptoms such as anxiety and depressive behavior [16,48]. Furthermore, PLX-PAD cells are considered to be hypoimmunogenic and have immunomodulatory properties: they reduce the production of the pro-inflammatory cytokines and induce secretion of the anti-inflammatory effects and by that facilitate healing of damaged tissue [32,49]. MSC therapy, like PLX-PAD cells, exhibits a new pharma-direction by identifying the specific injured cells among other tissue cells and rehabilitating them. This approach may minimize side effects and early drop-out [50]. Another benefit of the MSC therapy is the production of new active cells that integrate into the parenchyma and the neural circuit, which may reestablish functions lacking in cocaine addiction, thereby contributing to their amelioration.

To conclude, this research demonstrates that PLX-PAD cells attenuate cocaine-seeking behavior, likely through their capacity to migrate to specific mesolimbic regions and thereby improve the regions’ plasticity by restoration of neurons in the hippocampus. Moreover, this research suggests a noninvasive and non-repetitive treatment with an immediate effect and is maintained in the long term. We propose that cell therapy, specifically using PLX-PAD cells, could serve as a potential novel approach for attenuating cocaine craving. We postulate that IN administration of cell therapy is a non-invasive, safe, effective, novel treatment approach for addiction and may offer efficient rehabilitation.

### Forward-Looking Potential Improvements to the Therapeutic Method Proposed

Cell therapy and gene delivery are two promising therapeutic fields. It is tempting to combine them both for robust results. Gene delivery transmits the desired gene into a target cell or tissue via a vector. To accomplish a sufficient gene transmission for a certain time course, this vector, viral or non-viral, is required to be safe and effective. Non-viral methods, such as plasmid DNA application, are commonly considered safer for the host. Nevertheless, their efficiency is much less in vivo [51]. Genetic cell engineering faces critical problems in obtaining effective results and risk minimizations. Hence, shifting to clinical trials is challenging. Furthermore, although viral transfection methods exhibited high competence skills to express the target transferred gene, it has risk limitations such as chromosomal integration and instability, insertional mutagenesis, and proto-oncogene activation. Other methods, such as self-inactivating lentiviral vectors and a nonviral transfection transposon system, were suggested for MSC engineering; however, they posed safety risks insertional mutagenesis. Therefore, in the case of MSCs, a DNA-based transfection, which does not lead to genomic integration, is a much more desirable method. However, more work should be done to improve its low effectiveness; thus, later, it could become an efficient, effective, and safe tool [52]. NucleofectionTM application was a suggested [53] candidate for non-viral gene delivery in MSCs. Yet, further studies should be done to provide a cell line final product, preserve their optimal naïve MSC features, and promote an effective and non-risk therapeutic efficacy.

We used a selecting method to find specific PLX-PAD cells that may challenge pathogenic tissue to facilitate activation of specific gene expression that will improve a host’s microenvironment plasticity, thereby changing behavioral performance and leading to rehabilitation. This may be by secreting various neurotrophic factors such as BDNF and miRNAs (data not shown) that promote endogenous neurogenesis. Notably, the five main obstacles designated with the use of tissue-derived MSCs [54] may be overcome using our approach: 1. Shortages of tissue sources [38]—PLX-PAD cells derive from obtainable placenta [32,34,37]. 2. Cell population heterogeneity and low purity [38]—PLX-PAD cells are diagnosed with each tissue-purification batch, signifying a shelf product [32,34,37]. 3. Loss of pluripotency and capacities over continuous passages [38]—PLX-PAD cells are stable on use and do not differentiate or proliferate [34,37]. 4. Difficulty in using invasive [38] methods to affect the brain—We implant the PLX-PAD cells using IN. 5. Long-lasting efficacy [38]—Our recent results postulate normalization of the pathological plasticity, hence, long-lasting effect, upon the usage of PLX-PAD cells.

To sum up, using a theranostic approach may represent a future added potential of ideal shuttles of MSC-like cells pre-loaded with specific new and old anti-addictive agents (e.g., growth factors, micro RNAs), enabling affecting the plasticity of the pathological micro-environment, in parallel with local real-time monitoring of the therapeutic outcome.

## 5. Patents

The patent with Pluristem is: https://patentscope.wipo.int/search/en/detail.jsf?docId=WO2019021158&_cid=P10-L0GOQ9-64402-1, accessed on 5 May 2022.

## Figures and Tables

**Figure 1 pharmaceutics-14-01311-f001:**
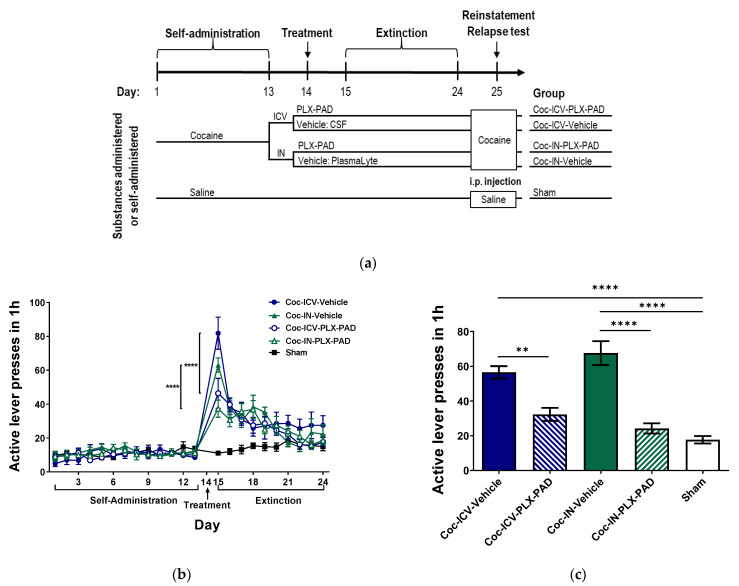
**Effect of PLX-PAD administered via intranasal (IN) and intracerebroventricular (ICV) routes on cocaine extinction and relapse of usage behavior in a self-administration model.** (**a**) **Self-administration model and experimental design**. Rats were placed daily for one hour into the self-administration (operant) chamber, where active lever presses were recorded. On Days 1–13 (self-administration phase), active lever presses resulted in cocaine injection. On Day 14 (treatment), rats were not placed in the operant chamber; they were randomized to PLX-PAD or Vehicle (CSF for ICV, PlasmaLyte for IN) treatment, which was administered ICV or IN. On Days 15–24 (extinction phase), rats were returned to the operant chamber daily with no cocaine available. On Day 25 (reinstatement test), rats were injected with cocaine (10 mg/kg. i.p.), then returned to the operant chamber with no further cocaine available. In the sham group, this protocol was conducted with saline instead of cocaine and without manipulation on Day 14; (**b**) **Effect of PLX-PAD administered ICV and IN on cocaine-seeking during extinction.** A significant number of active lever-presses on the first day of extinction in the Coc-ICV-Vehicle group compared to the Coc-ICV-PLX-PAD group, and Coc-IN-Vehicle compared to the Coc-IN-PLX-PAD group. Sham group did not demonstrate any change in the active lever presses on the first day of the extinction; two-way ANOVA with repeated measures, Bonferroni’s multiple comparison post hoc; **** *p* < 0.0001; (**c**) **Effect of ICV and IN PLX-PAD administration on reinstatement**. On the first day there was a significant effect of active lever presses on Day 25 in the ICV and IN PLX-PAD treatment compared to Coc-ICV-Vehicle and Coc-IN-Vehicle on drug-seeking behavior in the reinstatement test. Sham group did not demonstrate any change in the active lever presses in the reinstatement test; ** *p* < 0.01, **** *p* < 0.0001, Tukey’s multiple comparisons test. **Overall**: Data show mean ± SEM.

**Figure 2 pharmaceutics-14-01311-f002:**
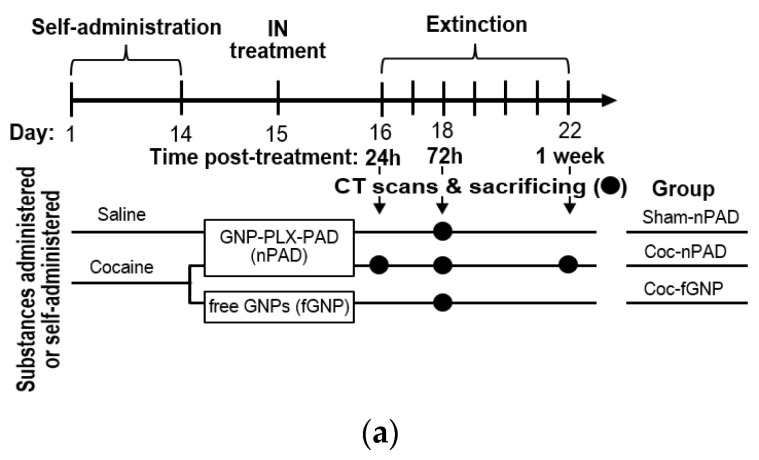
**PLX-PAD navigation in the brain.** (**a**) **Experimental design.** Rats underwent self-administration training with cocaine or saline on Days 1–14, then IN treatment on Day 15 with gold nanoparticle (GNP)-labeled PLX-PAD (nPAD) or with free GNPs (fGNP), and extinction training on Days 16–22. At 24 h, 72 h, and/or 1 -week post-treatment (as indicated). Rats from each group underwent in vivo CT scanning followed by sacrificing and analysis of brain regions, lungs, and liver by inductively coupled plasma optical emission spectrometry (ICP-OES). nPAD equivalents were estimated from GNP number by nPAD (see *GNP labeling* section) and fGNP groups; (**b**) **nPAD and fGNP accumulation in different brain regions 72 h post-treatment in relation to pathology.** GNPs in each brain region were quantified using ICP-OES (n = 4–7 per group). Significance is shown only for within-region comparisons. Significance of nPAD number in the PFC, NAc, and DG Ventricle brain regions in Cocaine+nPAD vs. Sham+nPAD (however, except for the DG and Ventricle regions, discussed in the Results and Discussion) and Cocaine+nPAD vs. Cocaine+fGNP; ** *p* < 0.01, **** *p* < 0.0001, Tukey’s multiple comparisons test. There was no significant main effect between groups in the Cb, BS, and OB brain regions; (**c**) **nPAD accumulation in reward-associated and -unassociated brain regions and peripheral organs, over time in Coc-nPAD group.** A comparison between Reward-associated brain regions (PFC, NAc, and DG), Reward-unassociated brain regions (Cb, BS, and OB), and Non-targeted organs (Liver and Lungs) n = 5–7 animals per group at each time point. Significance is shown only for within-time point comparisons; within-group differences across time points were not significant; ** *p* < 0.01, *** *p* < 0.001, **** *p* < 0.0001, Tukey’s multiple comparisons test; (**d**) **nPAD kinetics in reward-associated regions in cocaine-trained rats (Coc-nPAD).** n = 5–7 per data point. A post hoc Tukey showed in the DG, but not in the PFC and NAc, a significant difference in time point 24 h vs. time point 72 h. Contrarily, time point 24 h vs. time point 1-week and time point 72 h vs. time point 1-week showed no significant main effect; * *p* < 0.05, Tukey’s multiple comparisons test; **Overall**: Data show mean ± SEM.

**Figure 3 pharmaceutics-14-01311-f003:**
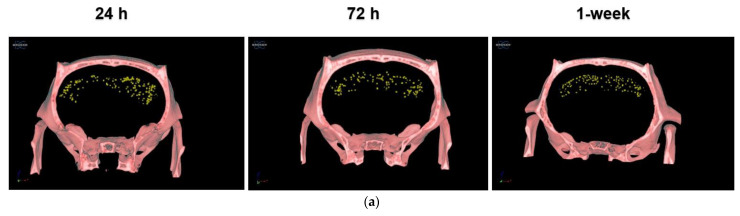
**Tracking PLX-PAD cells migration within the brain.** (**a**) **Representative in vivo micro-CT scans in the Coc-nPAD group.** Demonstrative micro-CT scans 24 h, 72 h, and 1-week post-IN treatment. Cell spreading into the brain is evident at 24 h and represented less in 72 h and 1 week; (**b**) **PLX-PAD cell localization in the dorsal DG 1-week post-treatment**. Representative images of the dorsal DG from Day 22, 1-week post-IN treatment show PLX-PAD localization; Scale bar: 50 μm. DG sections were stained with DAPI and Ku80 (a human-specific antibody recognizing PLX-PAD cells).

**Figure 4 pharmaceutics-14-01311-f004:**
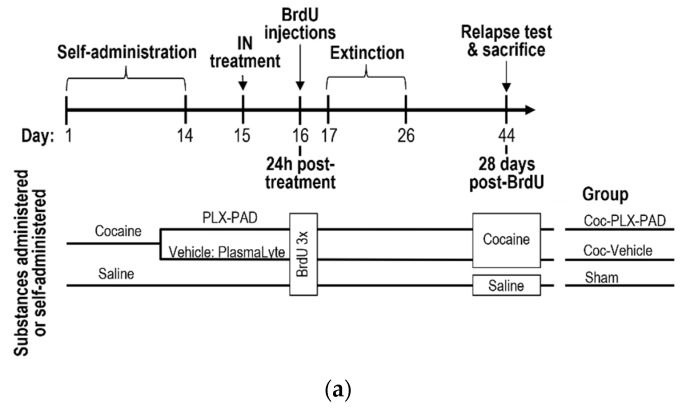
**Effect of PLX-PAD cells on neurogenesis in the hippocampus.** (**a**) **Experimental design.** Rats underwent cocaine or saline self-administration on Days 1–14, as in Figure 1. Cocaine-trained rats were treated IN with PLX-PAD or the vehicle (PlasmaLyte) on Day 15. On Day 16, all animals were injected with BrdU, a marker for neurogenesis (3 i.p injections at 4 h intervals). Animals underwent extinction training on Days 17–26. On Day 44, cocaine- and saline-trained animals were reinstated with cocaine or saline, respectively, and underwent the relapse test, as in Figure 1, after which rats were euthanized and perfused, and brains were excised. DG sections were stained for BrdU and NeuN, a marker of adult neurons; (**b**) **Adult neurogenesis in the dorsal DG.** Representative merged confocal microscope images of the dorsal DG from Day 44; ×:20; Scale bar: 50 μm, ×:10; Scale bar: 100 μm. Arrows show co-localization of BrdU and NeuN; (**c**) **Neurogenesis quantification: total BrdU-NeuN-positive cells in the DG.** There was a significant difference in the total BrdU-NeuN-positive cells counting in Coc-Vehicle vs. Coc-PLX-PAD groups; n = 4 per group * *p* < 0.05, Tukey’s multiple comparisons test; (**d**) **Effect of IN PLX-PAD administration on reinstatement.** There was a significant effect of active lever presses in the IN PLX-PAD treatment compared to Coc-IN-Vehicle on drug-seeking behavior in the reinstatement test. The sham group did not demonstrate any change in the active lever presses in the reinstatement test; * *p* < 0.05, ** *p* < 0.01, Tukey’s multiple comparisons (n = 5–6 per group); (**e**) **Correlation between drug-seeking behavior in the reinstatement test (panel** (**d**) **and neurogenesis (panel** (**c**). Data included only the n = 4 rats/group measured in (**c**). Scatter plot shows Spearman correlation coefficient r = −0.6063, * *p* = 0.0366. Overall: Data in panels (**c**,**d**) show mean ± SEM.

## Data Availability

The raw data supporting the conclusions of this article will be made available by the authors upon request.

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
