# Peer review of "Placenta-Derived Mesenchymal-like Adherent Stromal Cells as an Effective Cell Therapy for Cocaine Addiction in a Rat Model"

_pharmaceutics, 2022, doi:10.3390/pharmaceutics14071311_

Round 1
Reviewer 1 Report
All the reviewing comments were taken into account. I have no more critical remarks.
The only suggestion is to avoid abbreviations in the section subtitles -
"In vivo micro-CT scans" change to "In vivo micro computed tomography scans"
"PLX-PAD cells’ particular kinetic pattern in the DG" to "PLX-PAD cells’ particular kinetic pattern in the dentate gyrus"
Figures 2 and 3 have too large size - don't fit to one page, so the figure legend split between pages. It is not convenient to read. Might format figure panels on separate pages, make separate legends.
Overall, I recommend accept this work for publication now.
Reviewer 2 Report
Thank you for very nice job. I went through your manuscript there are few items need to be corrected:
Minors:
1-Line 344 nPad, should be changed to nPAD to keep homogeneity
2- Figure 2b the bar description (group) are missed.
Majors:
1-It would be great if you could show that overloading the gold on PLX-PAD cells does not change their phenotype or function (comparison with untouched cells).
2-Since you have been using human cell in healthy rat (without giving any immunosuppressive), it would be interesting to explain the immunogenicity of infused cells, inflammation in recipient’s body as well as immune reaction in the brain (although we do not expect much, but you were talking about cell leakage to liver and lung! Immune system reaction to the xenogeneic cell application is missed in your work.
